# A NOVEL APPROACH REVEALS HIGH ZOOPLANKTON STANDING STOCK DEEP IN THE SEA

Alexander Vereshchaka[1], Galina Abyzova[1], Anastasia Lunina[1], Eteri Musaeva[1], Tracey T. Sutton[2]

[1] Institute of Oceanology, Russian Academy of Sciences, Nakhimov Pr. 36, Moscow, 117997 Russia. +07 499-124-7940

[2] Halmos College of Natural Sciences and Oceanography, Nova Southeastern University, Dania Beach, FL 33004, United States. +01 954-262-3692

keywords: zooplankton communities, biological resources, decapods, vertical zones, plankton groups, vertical distribution

## ABSTRACT

In a changing ocean there is a critical need to understand global biogeochemical cycling, particularly regarding carbon. We have made strides in understanding upper ocean dynamics, but the deep ocean interior (> 1000 m) is still largely unknown, despite representing the overwhelming majority of Earth's biosphere. Here we present a method for estimating deep-pelagic zooplankton biomass on an ocean-basin scale. In so doing we have made several new discoveries about the Atlantic, which likely apply to the World Ocean. First, multivariative analysis showed that depth and Chl were the main factors affecting the wet biomass of the main plankton groups. Wet biomass of all major groups except fishes was significantly correlated with Chl. Second, zooplankton biomass in the upper bathypelagic domain is higher than expected, representing an inverted biomass pyramid. Third, the majority of this biomass comprises macroplanktonic shrimps, which have been historically underestimated. These findings, coupled with recent findings of increased global deep-pelagic fish biomass, revise our perspective on the role of the deep-pelagic fauna in oceanic biogeochemical cycling.

**INTRODUCTION**

The deep sea accounts for nearly 99% of the habitable volume of the planet (Dawson, 2012). Waters below 200 m are highly heterogeneous in space and time, harbouring diverse biological resources which are not yet quantitatively estimated. These ecosystems are and will continue to be impacted by climate change due to the cumulative effect of different stressors on their biota, including expanding oxygen minimum zones, shoaling of aragonite saturation horizons, acidification and warming (Okey *et al*., 2012). It is urgent that we estimate the biomass of the deep-sea biota for inventory purposes and for monitoring its changes in the future.

Studies on the deep-sea plankton biomass at selected sites include those in the North Pacific (e.g., Vinogradov, 1968; Murano et al., 1976; Yamaguchi et al., 2002ab; Yamaguchi, 2004) and Eastern Tropical Pacific (Sameoto, 1986), North Atlantic (Koppelmann and Weikert, 1992; 1999; Gislason, 2003; Vinogradov, 2005) and Mediterranean Sea (Scotto di Carlo et al., 1984; Weikert and Trinkaus, 1990), Indian Ocean (Vinogradov, 1968) and Arabian Sea (Koppelmann and Weikert, 1992; Böttger-Schnack, 1996). Fewer results concern deep-sea zooplankton distribution over larger areas (Longhurst & Williams, 1979; Gaard *et al*., 2008). The data regarding quantitative distribution of the deep-sea zooplankton for the Equatorial Atlantic and the South Atlantic Gyre are lacking. In addition to geographic restrictions, most deep-sea research has been concentrated on specific taxonomic groups (e.g. crustacean zooplankton; Burghart *et al*., 2007; Gaard *et al*., 2008), functional groups (e.g. gelatinous zooplankton; Lindsay & Hunt, 2005), or selected vertical zones (e.g. mesopelagic; Robison *et al*., 2010; Sutton *et al*., in press). Attempts to assess an entire deep-sea community have been rare and local (Vinogradov *et al*., 1996; Vereshchaka & Vinogradov, 1999; Vinogradov *et al*., 2000). Comparative assessments of entire water column plankton over large areas are absent.

Thus, it is timely to provide estimates of the zooplankton biomass throughout the water column over large areas. As any field data of the deep-sea zooplankton are inevitably local, we should find an indicator that is correlated with elements of the deep-sea zooplankton and that can be

assessed over large water areas/volumes. Here we offer and test a hypothesis that the zooplankton wet biomass in the deep-pelagic is correlated with surface chlorophyll concentration. This hypothesis has been corroborated for the epipelagic (0-200 m) layer, where correlations have been obtained (Vinogradov et al., 1999). It remains completely unknown, however, if this dependence is valid for the deep sea below 200 m. In theory, the standing stock of zooplankton should remain correlated with surface productivity and the correlation should decrease with depth. No large-scale data, however, are available on this subject. Here we attempt to fill that void by examining the relationship between remotely sensed sea surface data and *in situ*, discrete depth sampling data across the majority of the Atlantic Ocean (Fig. 1). In order to start this process, we will focus on the deep-sea meso- and macroplankton (1-10 cm length). This size fraction links primary and higher levels of oceanic production and is representatively sampled by the largest spectrum of plankton nets. As an indicator of surface productivity, surface chlorophyll concentration (Chl hereafter) derived from satellite information has been chosen as our indicator metric. We will check the presence of correlation for major groups of the zooplankton and for the different depth zones: epipelagic, main thermocline, upper- and lower-bathypelagic zones. If correlations exist, we will assess the standing stock of the plankton over vertical zones and over geographical areas. Where possible, we will estimate the role of major plankton groups and different depth zones in the total standing stock of the zooplankton. If successful, this attempt will provide a new expedient method for evaluation of deep-sea resources.

Zooplankton distribution is strongly affected by the presence of land (islands, continents, seamounts) and the sea-floor (Vereshchaka, 1995). The effect is prominent at a distance of tens of kilometer in the horizontal direction (Vereshchaka, 1990ab, 1994; Melo et al., 2014) and hundreds of meter in the vertical direction (Vereshchaka; 1995; Vereshchaka & Vinogradov, 1999; Cartes et al., 2010). In order to minimize the land and the sea-floor effects, this survey of the pelagic zooplankton in the open ocean will be made as far as possible from the bottom in the vertical direction and from the land in the horizontal direction.

**METHODS**
Field data were taken in the deep Central, South, and North Atlantic between 1996-2012 from
ultraoligotrophic to mesotrophic areas roughly between 40° S and 40° N during 36th and 37th
cruises of the R/V "Akademik Sergey Vavilov" (ASV), and 34th, 37th, 39th, 42nd, 46th, 47th, 49th,
50th cruises of the R/V "Akademik Mstislav Keldysh" (AMK - Table 1, Fig. 1). These areas
include the two main Atlantic Gyres (North and the South) and the Equatorial Atlantic between
them.
The whole database of this work contains two different datasets: (1) data of 2012-2013 (R/V
"Akademik Sergey Vavilov", mainly Central and South Atlantic) and (2) data of 1994-2005 (R/V
"Akademik Mstislav Keldysh", mainly North Atlantic). Samples have been taken with the same
protocol, but identification was much more precise for the first dataset. The community
composition, diversity, and other community patterns have been analyzed in detail for the first
dataset and presented in a recent paper (Vereshchaka et al., 2016). The second dataset contains
representative biomass values and significantly contributes to the metadata concerning deep
zooplankton; here we combine both datasets for a more comprehensive analysis. We excluded
data from temperate waters where the major spring peaks in primary production are being
exported from the euphotic zone (0–200 m depth) and reaching abyssal depths (4000 m) with a
significant time lag (e.g., 42 days: Smith et al., 2002); this lag differs for different depth zones
that may corrupt possible correlations.
Samples were taken between one hour after sunset and one hour before sunrise in order to make
a unified nighttime picture of the vertical distribution of animals. This method was adopted to
avoid the confounding effects of diel vertical migration. We sampled four discrete depth strata:
(1) the epipelagic zone (0-200 m), (2) the main thermocline (from 200 m to the depth of the 7°C
isotherm, within 550-800 m), (3) the zone from the lower boundary of the main thermocline to
1500 m, mainly Antarctic Transitional Waters, which we define here as the upper bathypelagic,
and (4) the layer 1500-3000 m, mainly North Atlantic Deep Waters, which we define here as the
lower bathypelagic (Fig. 2). The upper boundary of the bathypelagic zone as defined here did not
coincide with the traditional one (1000 m), because our sampling was associated with water
masses. The lower boundary of the bathypelagic was 3000 m instead of usually adopted 4000 m,
as we had to avoid sampling of the benthopelagic zone.
We used a closing Bogorov-Rass (BR) plankton net (1-m$^2$ opening, 500-µm mesh size, towed at
a speed of 1 m sec$^{-1}$), which was proven to successfully sample deep-sea plankton $\geq$ 1.0 mm long
(Vinogradov *et al*., 1996; 2000); smaller animals may pass through the sieve during filtration.
The net was deployed at the maximal depth of haul, then opened and towed vertically upwards,
and finally closed at the minimal depth of haul with a mechanical device. The minimal
horizontal distance between station and the land was 400 km and the minimal vertical distance
the lower boundary of the deepest haul and the sea-floor was 750 m (Table 1), so that the
land/sea-floor effect could be ignored.
We divided the net plankton into four major groups: non-gelatinous mesozooplankton (mainly
copepods and chaetognaths; 1-30 mm length), gelatinous mesozooplankton (mainly
siphonophorans and medusae; individual or zooid; 1-30 mm length), decapods and small
(macroplanktonic) fishes (both groups over 30 mm length). Identification was done according to
literature (e.g., Rose, 1933; Brodsky, 1950; Mauchline & Fisher, 1969; Brodsky *et al*., 1983;
Markhasheva, 1996). Synonymy of species was corrected according to www.marinespecies.org.
Decapods, fishes, and gelatinous species were weighed with a precision of 0.1 g before fixation.
Wet weight of mesoplanktonic groups was estimated according to adopted procedures
(Vinogradov *et al*., 1996; 2000; Gaard, E., *et al*., 2008). In brief, wet weight $w_{tot}$ of the non-
gelatinous mesozooplankton (mainly copepods) was estimated as $w_{tot} = \Sigma\ (k\ *\ l_i^3)$, where $l_i$ is
length of an individual specimen, $k$ is a species-dependent coefficient; tables of these coefficients
have been published elsewhere (e.g., Vinogradov & Shushkina, 1987).
Surface chlorophyll-a concentration (Chl) derived from satellite images was used as a measure of
the surface productivity. Chl data were taken from Aqua MODIS (level 3, 4-km resolution) from
2003 to 2015. Before this period Chl data were taken from SeaWiFS (level 3, 9-km resolution)
from 1997 to 2002. Chl data were averaged over one year preceding the sampling date and over
a 5º $\times$ 5º square (with the sampling site in the center).
In order to establish relationships between the major plankton group wet biomass and possible
environmental factors, Canonical Correspondence Analysis (CCA: Ter Braak, 1986) was
performed on major group biomass using an assortment of environmental variables: temporal
(month and year), spatial (latitude, longitude, and depth), and surface chlorophyll concentration
(Chl). As the sampling was associated with distinct water masses, such environmental
parameters as temperature, salinity, and depth were correlated and only one of them, the depth,
was included in CCAs. CCA is a powerful multivariate technique to extract synthetic
environmental gradients from ecological data (Ter Braak and Verdonschot, 1995). Ordination
axes are based on the measured environmental variables and represent linear combinations of the
variables. Arrows showing variables in the ordination plots are proportional in length to the
importance of each variable (Ter Braak, 1986), and therefore community variation can be
directly related to environmental variation. CCAs included either all hauls, or hauls from
separate strata and made possible to assess the contribution of all analyzed factors.
Calculations, statistical procedures, regression analysis, an ANOVA tests were carried out with
the use of Excel and STATISTICA, CCAs with PAST 3.04 (Hammer et al., 2001).

**RESULTS**
Over 300 taxa were identified, counted, measured, and their weight calculated to estimate
standing stocks (the plankton assemblages are considered in detail elsewhere - Vereshchaka et
al., 2016). The main contribution to the total zooplankton standing stock was made by decapod
decapods, followed by non-gelatinous mesozooplankton, gelatinous mesozooplankton, and fishes
(Table 2).
The epipelagic zone was dominated by the two groups of mesozooplankton, the main
thermocline was dominated by non-gelatinous mesozooplankton and decapods, the upper
bathypelagic zone was dominated by decapods, and the lower bathypelagic zone was dominated
by gelatinous zooplankton (Table 2). The dominant role of decapods will be further quantified as
a separate parameter, the share of decapods in the total plankton wet biomass (%).
Actual vertical distribution of major groups varied, but typical profiles are represented for the
northwest and northeast corners of studied area (Fig. 3 AB), for the central part and the eastern
periphery of the North Atlantic Gyre (Fig. 3 CD), and for the Equatorial area and southwestern
periphery of the South Atlantic Gyre (Fig. 3 EF).
Multivariative CCA with all hauls included (Fig. 4A) showed aggregation of hauls in two
groups. The first group (the left of OY axis) was mainly represented by the epi- and lower
bathypelagic hauls and related to non-gelatinous, gelatinous, and total plankton. The second
group (the right of OY axis) was represented by the and upper/lower bathypelagic hauls and
related to the share of decapods. The first factor (F1) was mainly linked to depth, the second
factor (F2) was primarily associated with Chl (Fig. 4A). Contribution of other factors was less
significant. Such variables as Chl and depth had the largest effect on wet biomass of all major
groups, the share of decapods was mostly linked to depth.
Multivariative CCA with only epipelagic hauls (Fig. 4B) showed one group of samples. The first
factor (F1) was mainly linked to Chl, the second factor (F2) was primarily associated with month
(Fig. 4B). Chl had the largest effect on biomass of both mesoplanktonic groups and total
plankton, decapods and fish were also linked to month.
Multivariative CCA with hauls from the main thermocline (Fig. 4C) showed aggregation of
hauls in two groups: one was mainly related to fishes and the share of decapods (the left of OY
axis), another was linked to both groups of mesoplankton and total plankton (the right of OY
axis). The first factor (F1) was mainly linked to year and latitude, the second factor (F2) was
primarily associated with longitude (Fig. 4C).
Multivariative CCA with upper bathypelagic hauls (Fig. 4D) showed aggregation of hauls in two
groups: one was mainly related to the share of decapods (the left of OY axis), another was linked
to main plankton groups (the right of OY axis). The first factor (F1) was mainly linked to Chl,
the second factor (F2) was primarily associated with month and year (Fig. 4D).
Multivariative CCA with lower bathypelagic hauls (Fig. 4E) showed aggregation of hauls in two
groups: one was mainly related to the share of decapods (the left of OY axis), another was linked
to plankton groups (the right of OY axis). The first factor (F1) was mainly linked to longitude
and year, the second factor (F2) was primarily associated with Chl (Fig. 4E).
Multivariative CCA with wet biomass values integrated over whole water column (Fig. 4F)
showed aggregation of hauls in two groups: one was mainly related to the share of decapods (the
right of OY axis), another was linked to plankton groups (the left of OY axis). The first factor
(F1) was mainly linked to Chl, month, and year, the second factor (F2) was primarily associated
with geographical coordinates (Fig. 4F).
Results of multivariate analyses allow search for possible correlations between wet biomass of
the major plankton groups and the most important environmental factor, Chl. The total
zooplankton wet biomass in the whole water column and the biomass of all major faunal groups
except fishes were highly correlated with the averaged Chl (Fig. 5, Table 3). Moreover, in most
cases the standing stock of the major groups except fishes in each of the vertical zones was also
correlated with Chl; the dependence was more robust for upper vertical zones and weakened
with depth. Fish wet biomass was never robustly correlated with Chl.
Having the correlation between the total zooplankton standing stock and Chl, we calculated the
total zooplankton standing stock (wet biomass under 1 $m^{-2}$ in the whole water column) and
standing stocks within each strata (wet biomass under 1 $m^{-2}$ integrated over whole layer) over
selected areas. We did that for three rectangular areas roughly corresponding to the North and
South Atlantic Gyres and the Equatorial Atlantic (Fig. 6). The maximum plankton stock was
found in the Equatorial Atlantic ($3.8 \times 10^7$ t wet weight), with the South and North Gyres being
approximately half ($2.2 \times 10^7$ t) and one-quarter ($1.0 \times 10^7$ t) of this amount, respectively.
Contribution of various vertical zones to the total plankton standing stock was similar in the
three selected areas (Fig. 6). The contribution of the main thermocline zone was the smallest
portion of the total plankton stock (13-16 %), the epipelagic and lower bathypelagic zones were
intermediate (15-25 %), and the upper bathypelagic zone contributed the highest portion (41-48
%). In terms of faunal contributions, gelatinous and non-gelatinous mesozooplankton accounted
for nearly one-quarter of the total zooplankton stock, while the decapods composed
approximately half. Various species of the decapod genera *Acanthephyra* A. Milne-Edwards,
1881 and *Gennadas* Spence Bate, 1881 were dominant throughout the studied area, *Notostomus*
A. Milne-Edwards, 1881 and *Systellaspis* Spence Bate, 1888 were dominant in the Equatorial
area and South Atlantic Gyre. Fishes (represented by Gonostomatidae Cocco, 1838 and
Myctophidae Gill, 1893) were not included in this analyses, since their biomass was not
correlated with studied environmental parameters.

**DISCUSSION**
Although scant on the global scale, our deep-sea samples collected during the last 20 years using
standardized methods throughout the whole water column provide an unprecedented opportunity
to investigate the distribution of zooplankton biomass at an ocean-basin scale. This is the first
snapshot of the biomass distribution throughout the whole water column over a significant
oceanic area. Further, this is a first attempt to quantitatively connect the dots related to surface
productivity and deep-sea zooplankton biomass, including the bathypelagic zone, which
contained the highest portion of water column meso/macrozooplankton standing stock.
The wet biomass profiles (Fig. 3), although different at various sites, show same
quasiexponential decrease of the mesoplankton biomass, as has been known before (e.g.,
Vinogradov, 1970). As for novelty, high decapod biomasses are recorded from many sites. Since
these animals may avoid plankton nets, high biomass values are even more striking. Our data do
not allow detailed analysis of profiles, because vertical resolution of samples is lower than
necessary, but assessment of factors influencing biomass values is possible.
Multivariate analysis showed that depth and Chl were the main general factors affecting the
wet biomass of main plankton groups (Fig. 4A). Obtained regressions between Chl and biomass
of the major plankton groups are obfuscated by several factors. First, algorithms for conversion
of satellite images to Chl data are not perfect (Watson *et al*., 2009). Second, Chl data, even if
estimated unerringly, do not reflect surface productivity thoroughly: autotrophic organisms may
live far below the surface and even create deep maxima with significant chlorophyll
concentration not detectable via satellites (Uitz *et al*., 2006). Third, the trophic structure of deep-
pelagic communities and deep-water circulation locally differ, thus providing different

conditions for downward energy transfer and accumulation of organic matter in the zooplankton wet biomass. It is all the more interesting that our data do show statistically significant correlation between Chl and the deep zooplankton biomass. The use of Chl averaged over $5^\circ$ x $5^\circ$ area and one-year period provide a new and productive approach to assess the deep pelagic biomass. The use of different temporal and spatial scaling may improve this approach in the future.

Although our results provide a means for calculating global zooplankton wet biomass by integrating satellite remote sensing with *in situ* sampling, some caveats must be noticed, including:

- Correlations may be different outside the tropical/subtropical region of the Atlantic Ocean. Studies in the epipelagic zone show that such correlations are better in warm waters than in the cold waters (Vinogradov *et al.,* 1999).

- Correlations may be different in different oceans. Our data show better correlation between the Chl concentration and the zooplankton wet biomass in the epipelagic zone than in Vinogradov *et al*. (1999) - 0.67 versus 0.53. We used field data from the Atlantic Ocean only, while Vinogradov *et al*. . (1999) based their studies on a set of data from the Atlantic, Indian, and Pacific Oceans. Each ocean probably requires an individual approach until conversion factors can be obtained to link geographically distant deep-sea assemblages.

- Actual wet biomass of gelatinous mesozooplankton is underestimated by our gear. A significant part of ctenophores and medusae are destroyed in the mesh during retrieval. Fragile gelatinous animals may dominate in the deep sea (Robison *et al*., 2010) and plankton nets are suboptimal for estimating their actual abundance (Vereshchaka & Vinogradov, 1999).

- Actual wet biomass of the decapods is also underestimated, as these animals likely avoid plankton nets and trawls to some extent (Vereshchaka, 1990).

Probably the most striking result we found was the unexpectedly high decapod wet biomass.
Macroplanktonic decapod biomass, even in the maximum layers, is typically 0.05-0.5 mg m$^{-3}$
and never exceeds 1.0 mg m$^{-3}$ in the Atlantic (Foxton, 1970a, b), Indian (Vereshchaka, 1994),
and in the Southeast Pacific (Vereshchaka, 1990). The values presented are one order of
magnitude higher (Table 1), which seems paradoxical, as the nets were smaller and should have
ostensibly caught fewer and smaller decapods. Our observations from submersibles show that
deep-sea decapods are generally stationary in the water column with abdomens oriented slightly
upward. When disturbed, decapods try to escape and jump upward using the abdomen and tail
fan. This behaviour is effective in the pelagic realm where predators are thought to attack from
below and thus many deep-pelagic decapods possess downward-oriented photophores for
counter-illumination (Widder, 1999). Upward jumps are also effective to escape from a net or a
trawl that is traditionally towed in the horizontal direction. The BR net, however, is towed
vertically and the decapods may have less chance to avoid the gear.
In contrast to decapods, pelagic fishes escape in horizontal direction, as has been observed from
submersibles many times by the authors. This reaction is successful when vertical hauls are used
and our results are thus not representative for assessment of the pelagic fish biomass. This
biomass may occur to be finally correlated with Chl but horizontally towed gears are necessary
to prove that.
The dominance of macroplanktonic decapods in the deep sea illustrates an inverted biomass
pyramid, as their biomass is larger than that of their prey (non-gelatinous mesozooplankton).
This happens because decapods (typical life spans of several years) grow and reproduce much
slower than mesozooplankton (typical life span several months), which equates to a low
production rate relative to its high standing stock; ergo, the energy pyramid is not inverted. Thus,
the decapod distribution offers additional example of the inverted biomass pyramid described for
plankton communities (Gasol *et al*., 1997).

The most significant contribution to the total zooplankton standing stock unexpectedly came
from the upper bathypelagic zone, not the epipelagic zone or the main thermocline (Fig. 6). The
upper bathypelagic zone was dominated by macroplanktonic decapods, which accounted for over
half of the standing stock wet biomass. Most decapods undertake diel vertical migration (Foxton,
1970a,b), feeding on mesozooplankton in the upper layers at night and hiding from predators in
the dark upper bathypelagic zone by day. This behaviour appears effective and provides good
prospects for biomass accumulation below the main thermocline in the ocean. The finding of
higher than expected biomass deep in the water column mirrors recent findings that suggest
deep-pelagic fish biomass has been underestimated by up to an order of magnitude (Kaartvedt *et*
*al*., 2012; Irigoien *et al*., 2014). The global ramifications of these findings, coupled with ours, are
that energy transfer efficiency from phytoplankton to intermediate and higher trophic levels in
oceanic ecosystems has been underestimated, and that both zooplankton and fishes are likely
respiring a large portion of the primary production in the deep-pelagic realm.

**ACKNOWLEDGEMENTS**
The studies were supported by the Presidium Programms 3P of the Russian Academy of
Sciences.

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

**Table 1. List of stations, cruises of R/V "Akademik Sergey Vavilov" (ASV) and R/V "Akademik Mstislav Keldysh" (AMK). Sampling zones: E - epipelagic, M - main thermocline, U- upper bathypelagic, L - lower bathypelagic; T - total haul (0-3000 m, net was not closed).**

| No of Station | Date | Latitude | Longitude | Sampling zones | Depth, m |
|---|---|---|---|---|---|
| 2474 ASV | 24.10.2012 | 9°25' N | 19°44' W | EMUL | 4282 |
| 2479 ASV | 25.10. 2012 | 3°51' N | 21°15' W | EMUL | 5235 |
| 2483 ASV | 28.10. 2012 | 0°50' N | 22°26' W | EMUL | 4360 |
| 2488 ASV | 29.10. 2012 | 6°12' S | 24°05' W | EMU | 3800 |
| 2489 ASV | 30.10. 2012 | 10°18' S | 26°37' W | EMUL | 5500 |
| 2490 ASV | 01.11. 2012 | 15°06' S | 28°45' W | EMUL | 5030 |
| 2491 ASV | 03.11. 2012 | 22°43' S | 32°05' W | EMUL | 4690 |
| 2492 ASV | 05.11. 2012 | 26°39' S | 33°58' W | EMUL | 4710 |
| 2498 ASV | 07.11. 2012 | 29°27' S | 39°15' W | EMUL | 4724 |
| 2499 ASV | 10.11. 2012 | 32°11' S | 46°26' W | T | 3780 |
| 2500 ASV | 23.09. 2013 | 41°58' N | 14°17' W | EMUL | 5000 |
| 2504 ASV | 27.09. 2013 | 31°12' N | 20°48' W | EMU | 3150 |
| 2505 ASV | 29.09. 2013 | 26°14' N | 21°03' W | EMUL | 4700 |
| 2506 ASV | 30.09. 2013 | 19°59' N | 21°22' W | EMUL | 3780 |
| 2507 ASV | 03.10. 2013 | 11°50' N | 21°47' W | EMUL | 4900 |
| 2508 ASV | 04.10. 2013 | 5°50' N | 22°00' W | EMUL | 3800 |
| 2518 ASV | 10.10. 2013 | 1°25' S | 24°00' W | EMUL | 4700 |
| 2519 ASV | 11.10. 2013 | 07°01' S | 26°04' W | EMUL | 4500 |
| 2520 ASV | 14.10. 2013 | 15°35' S | 28°41' W | EMUL | 5100 |
| 2524 ASV | 19.10. 2013 | 26°23' S | 32°53' W | EMU | 3000 |
| 2528 ASV | 21.10. 2013 | 31°00' S | 40°38' W | EMU | 2250 |
| 3341 AMK | 12.09.1994 | 29°06' N | 43°12' W | EMUL | 3205 |
| 3365 AMK | 16.09. 1994 | 26°12' N | 44°54' W | EMUL | 3887 |
| 3604 AMK | 08.09. 1995 | 41°42' N | 49°54' W | EMUL | 3749 |
| 3671 AMK | 28.08. 1996 | 29°06' N | 43°12' W | EMUL | 5270 |
| 3816 AMK | 10.09. 1998 | 41°42' N | 49°54' W | EMUL | 3750 |
| 3854 AMK | 28.10. 1998 | 36°12' N | 33°54' W | EMU | 2470 |
| 3980 AMK | 9-10.10. 1999 | 36°12' N | 33°54' W | EMUL | 3285 |
| 4149 AMK | 10-11.06. 2001 | 48°06' N | 16°06' W | EMUL | 4700 |
| 4301 AMK | 01-04.06. 2002 | 48°06' N | 16°06' W | EMUL | 4800 |
| 4547 AMK | 25-26.06. 2003 | 41°42' N | 49°54' W | EMUL | 3700 |
| 4580 AMK | 30.07. 2003 | 37°54' N | 31°30' W | EMU | 2070 |
| 4601 AMK | 08.08. 2003 | 30°06' N | 42°06' W | EMU | 1800 |
| 4613 AMK | 12-13.08. 2003 | 23°24' N | 45° 00' W | EMUL | 4700 |
| 4791 AMK | 24-25.08. 2005 | 29°06' N | 43°12' W | EMU | 3070 |
| 4799 AMK | 28.08. 2005 | 30°06' N | 42°06' W | EMU | 2545 |

**Table 2. Average values ± standard deviation of wet biomass the major plankton groups in the whole water column (g m$^{-2}$) and vertical zones (mg m$^{-3}$) of the Atlantic Ocean.**

| Vertical zones | Non-gelatinous mesoplankton | Gelatinous mesoplankton | Decapods | Fishes | Total plankton | Number of samples |
|---|---|---|---|---|---|---|
| Whole water column (0-3000 m) | 13.38±24.08 | 8.07±17.33 | 15.63±31.04 | 1.25±2.32 | 37.08±58.49 | 36 |
| Epipelagic zone | 28.32±54.86 | 20.16±53.96 | 0.58±2.16 | 0.62±0.86 | 49.07±78.19 | 35 |
| Main thermocline zone | 5.68±12.34 | 1.86±4.03 | 5.40±9.26 | 0.38±0.63 | 12.93±18.53 | 35 |
| Upper bathy-pelagic zone | 4.30±9.20 | 4.12±11.14 | 12.07±25.73 | 0.61±0.81 | 20.49±36.28 | 35 |
| Lower bathy-pelagic zone | 0.19±0.16 | 1.79±4.40 | 0.04±0.16 | 0.04±0.16 | 2.02±9.71 | 26 |

**Table 3. Correlation between surface chlorophyll-a concentration (Chl, mg m$^{-2}$) and wet biomass (g m$^{-2}$ for the whole water column and mg m$^{-3}$ for vertical zones): coefficients of determination (R$^2$), equations, and levels of significance (\*\*\*\* indicate $p$ <0.001, \*\*\* for $p$ <0.01, \*\* for $p$ <0.01, \* for $p$ <0.05).**

| Vertical zones | Non-gelatinous mesoplankton | | Gelatinous mesoplankton | | Decapods | | Fishes | | Total plankton | |
|---|---|---|---|---|---|---|---|---|---|---|
| | R$^2$ | Regression equations | R$^2$ | Regression equations | R$^2$ | Regression equations | R$^2$ | Regression equations | R$^2$ | Regression equations |
| Whole water column, n=36 | 0,5129\*\*\*\* | B = 121,54Chl - 6,3663 | 0,1971\*\*\* | B = 54,22Chl - 0,7371 | 0,1609\*\* | B = 87,747Chl + 1,3743 | 0,026 | B = 2,5959Chl + 0,8296 | 0,4235\*\*\*\* | B = 266,1Chl - 4,8996 |
| Epipelagic zone, n=35 | 0,4674\*\*\*\* | B = 258,84Chl - 13,001 | 0,1374\*\*\* | B = 138,03Chl - 1,8744 | 0,028 | B = 2,6517Chl + 0,181 | 0,0151 | B = -0,7718Chl + 0,7576 | 0,544\*\*\*\* | B = 397,96Chl - 13,863 |
| Main thermocline zone, n=35 | 0,4082\*\*\*\* | B = 53,63Chl - 3,2029 | 0,2971\*\*\*\* | B = 14,929Chl - 0,6162 | 0,0512 | B = 15,151Chl + 3,0384 | 0,0012 | B = -0,156Chl + 0,418 | 0,3925\*\*\*\* | B = 78,963Chl + 0,0833 |
| Upper bathy-pelagic zone, n=35 | 0,4152\*\*\*\* | B = 40,335Chl - 2,3795 | 0,0569 | B = 18,065Chl + 1,1344 | 0,2118\*\* | B = 85,558Chl - 1,2733 | 0,0216 | B = -0,8529Chl + 0,7653 | 0,2599\*\*\*\* | B = 125,8Chl - 0,0726 |
| Lower bathy-pelagic zone, n=26 | 0,284\*\*\* | B = 14,61Chl - 0,9625 | 0,1518\* | B = 2,6226Chl + 0,0143 | 0.1263\*\* | B = 22,622Chl + 0,7622 | 0,0484 | B = 0,2334Chl + 0,0038 | 0,1715\* | B = 34,942Chl + 0,3334 |

Figure Legends

Figure 1.  Deep-sea plankton stations (black circles) sampled during the cruises of R/V "Akademik Sergey Vavilov" (ASV) and R/V "Akademik Mstislav Keldysh" (AMK) (see also Table 1). Background: surface chlorophyll-a concentration averaged over 2013, scale (mg m$^{-2}$) on right.

Figure 2. Temperature (°C, left) and salinity (‰, right) along the transect A16 (Koltermann et al., 2011).

Figure 3. Wet biomass profiles (mg m$^{-3}$) of the main plankton groups in the epipelagic (1), main thermocline (2), upper bathypelagic (3) and lower bathypelagic obtained during the cruises of R/V "Akademik Sergey Vavilov" (ASV) and R/V "Akademik Mstislav Keldysh" (AMK) (see also Fig. 1 and Table 1).

Figure 4. CCAs of all hauls included (A), of hauls taken in the epipelagic (B), main thermocline (C), upper- (D) and lower bathypelagic (E), and of the calculated standing stocks (F). Two first axes (F1 and F2) with respective explained variance represented.

Figure 5. Wet biomass of major plankton groups (vertical axes) in the whole water column (g m$^{-2}$) and in different vertical zones (mg m$^{-3}$) versus surface chlorophyll (horizontal axes, mg m$^{-2}$).

Figure 6. The standing stock (wet biomass) of the deep-sea plankton and contribution (%) of vertical zones in the North, Equatorial, and South Atlantic. Background: surface chlorophyll-a concentration averaged over 2013, scale (mg m$^{-2}$) on right. Yellow circles: stations.

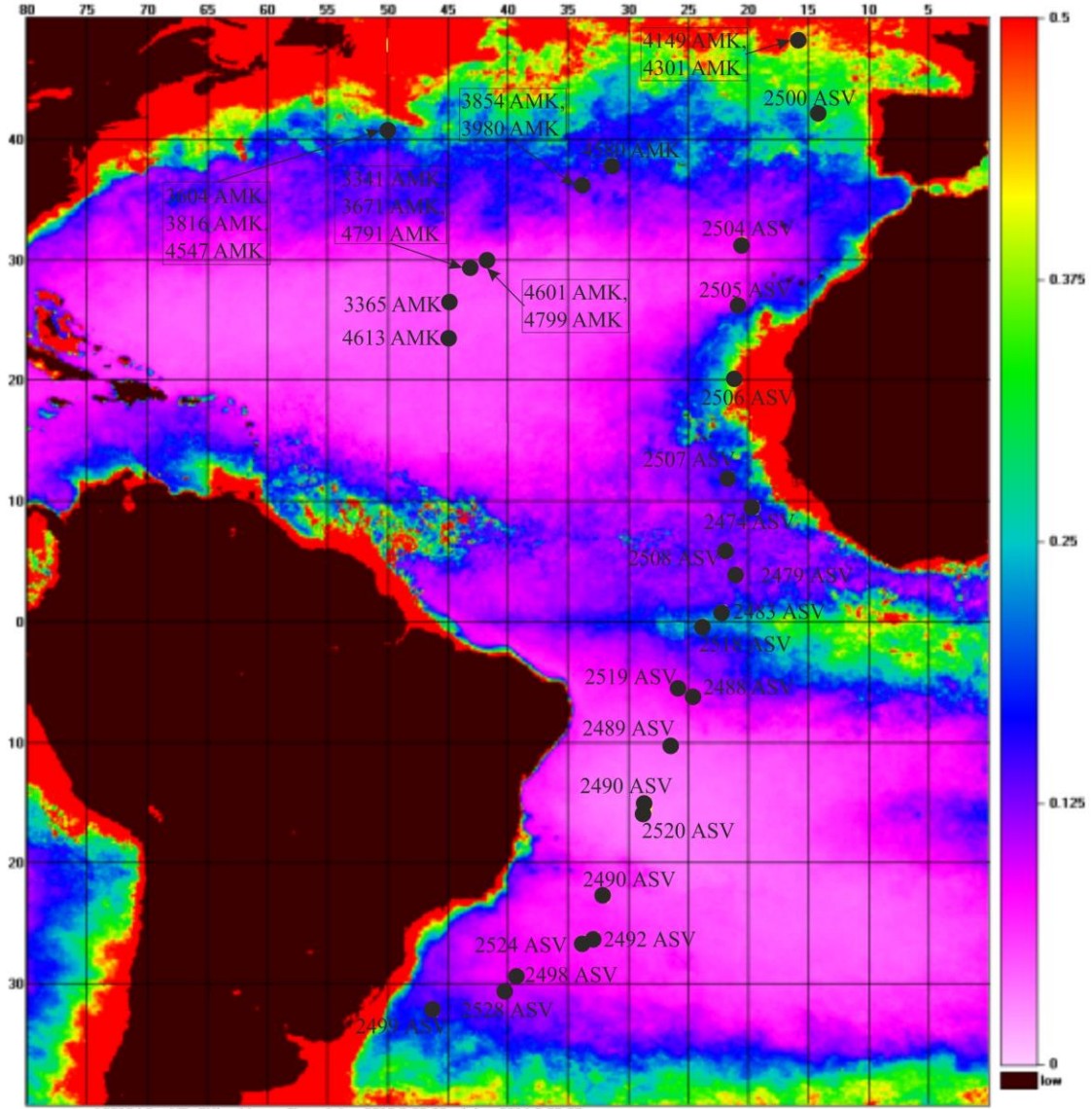

Figure 1.

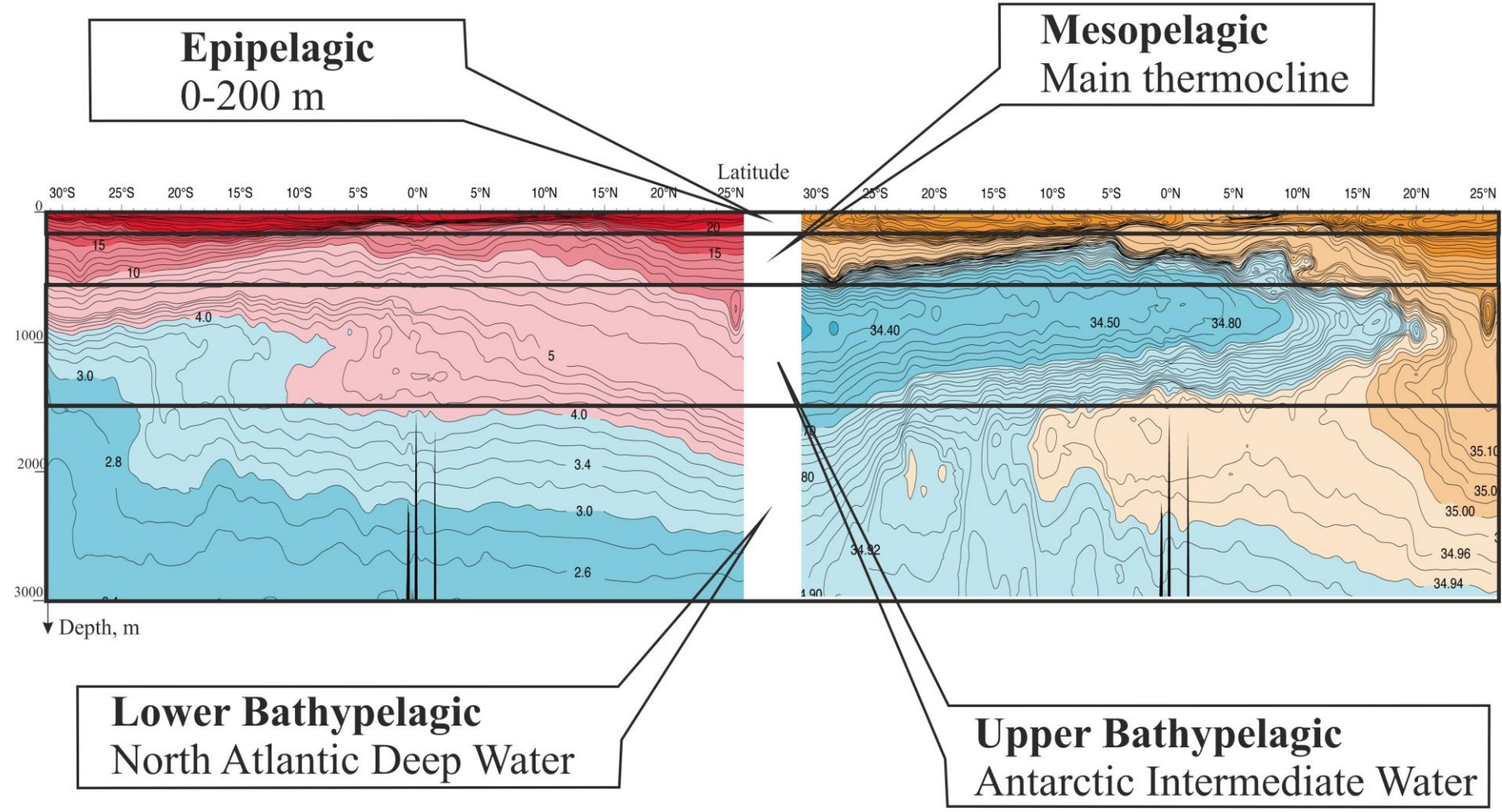

**Epipelagic**
0-200 m

**Mesopelagic**
Main thermocline

**Lower Bathypelagic**
North Atlantic Deep Water

**Upper Bathypelagic**
Antarctic Intermediate Water

Figure 2.

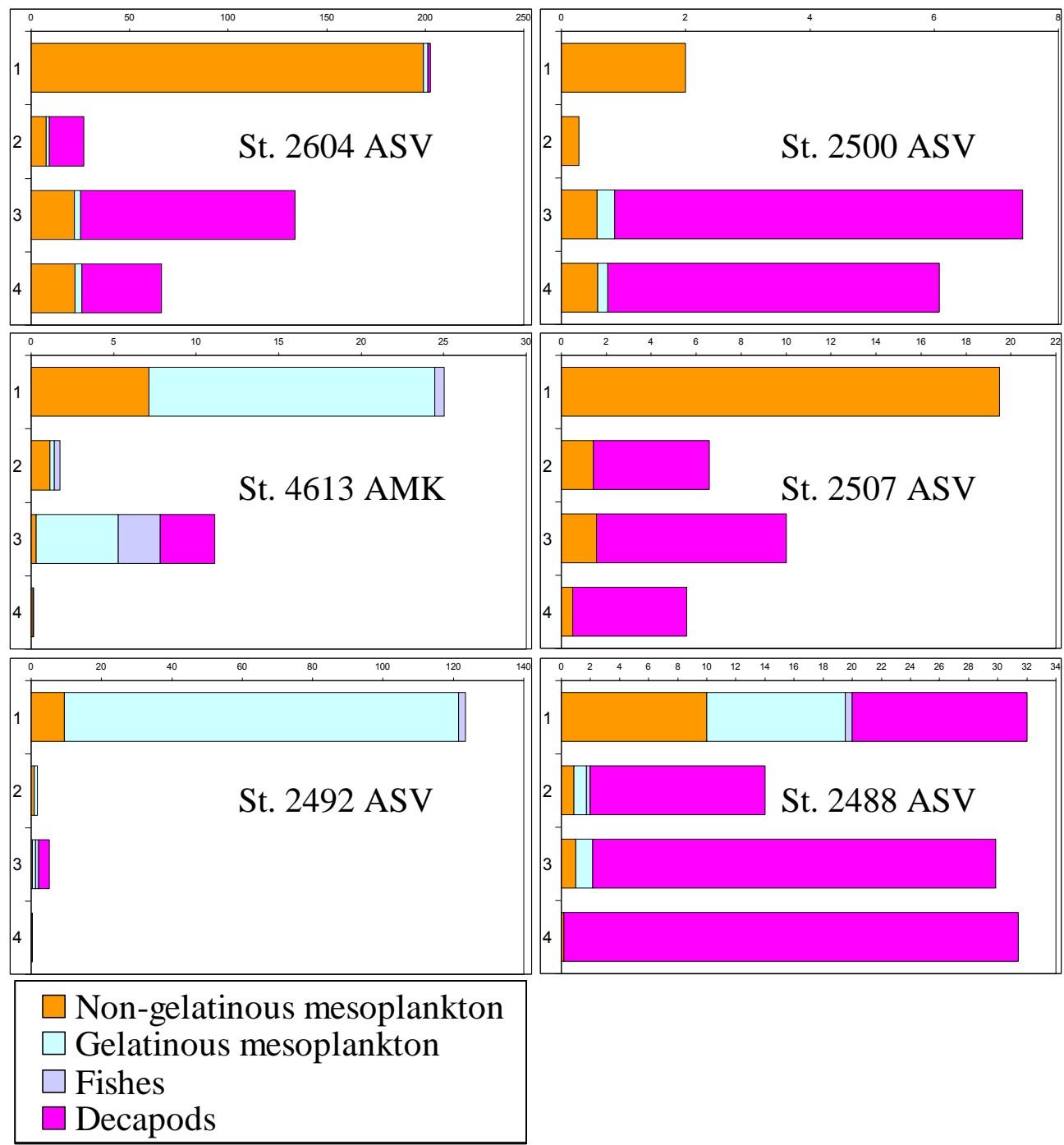

Figure 3.

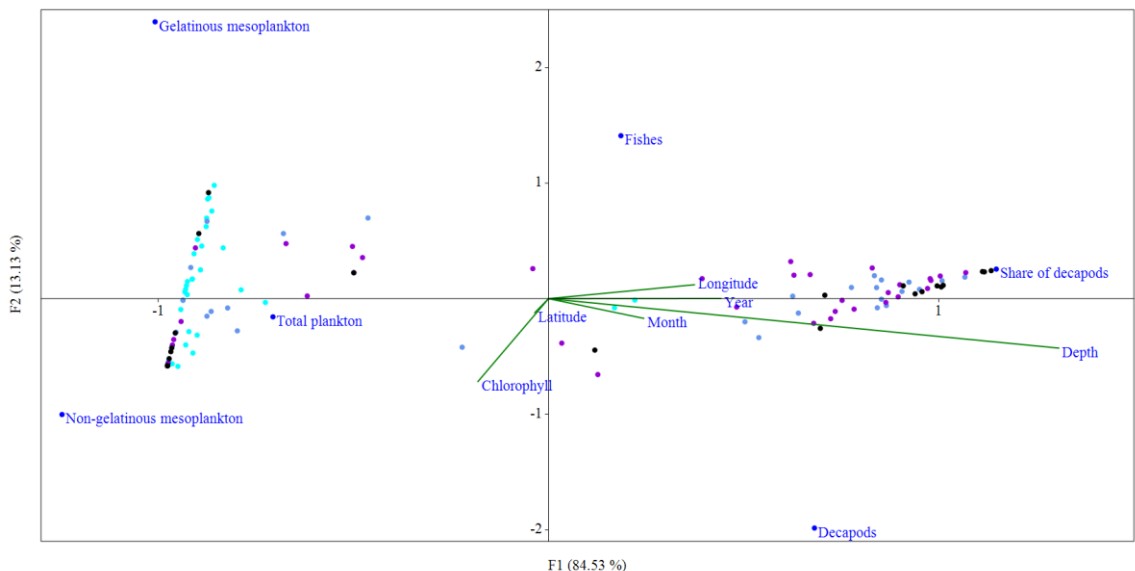

A

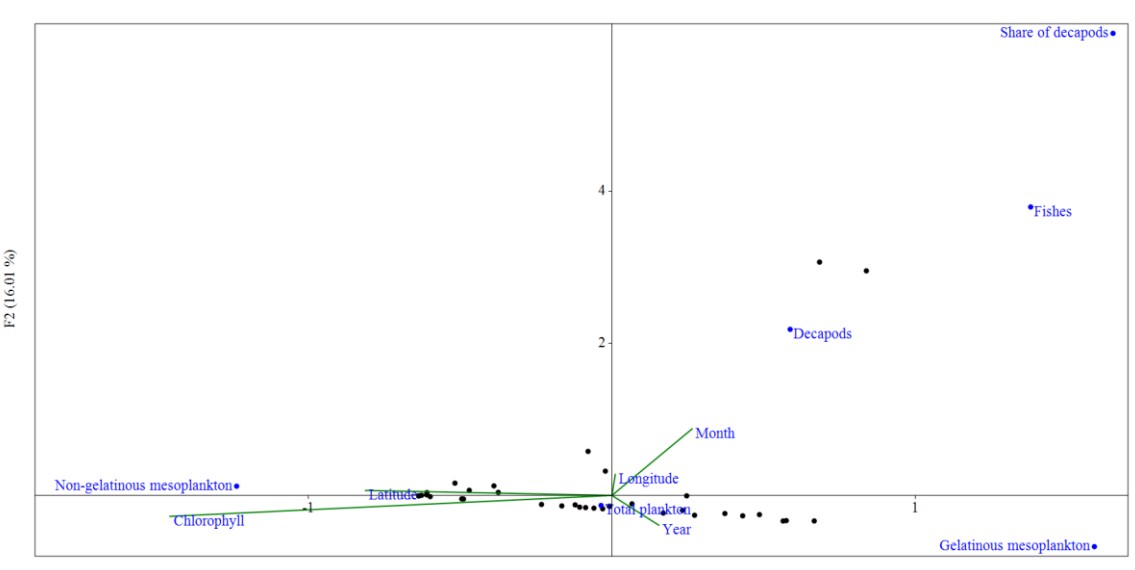

B

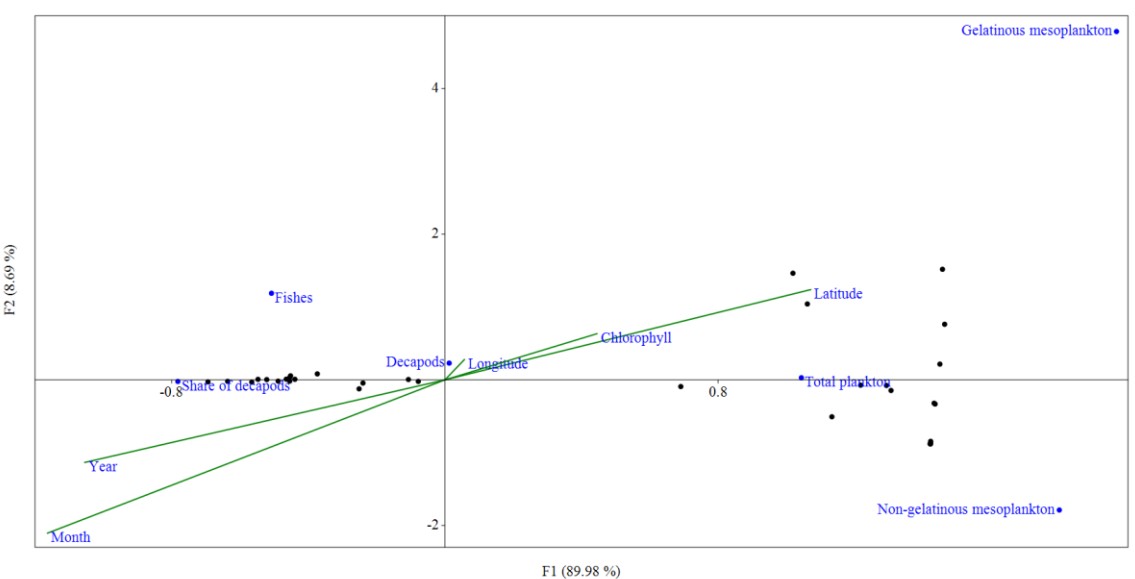

C

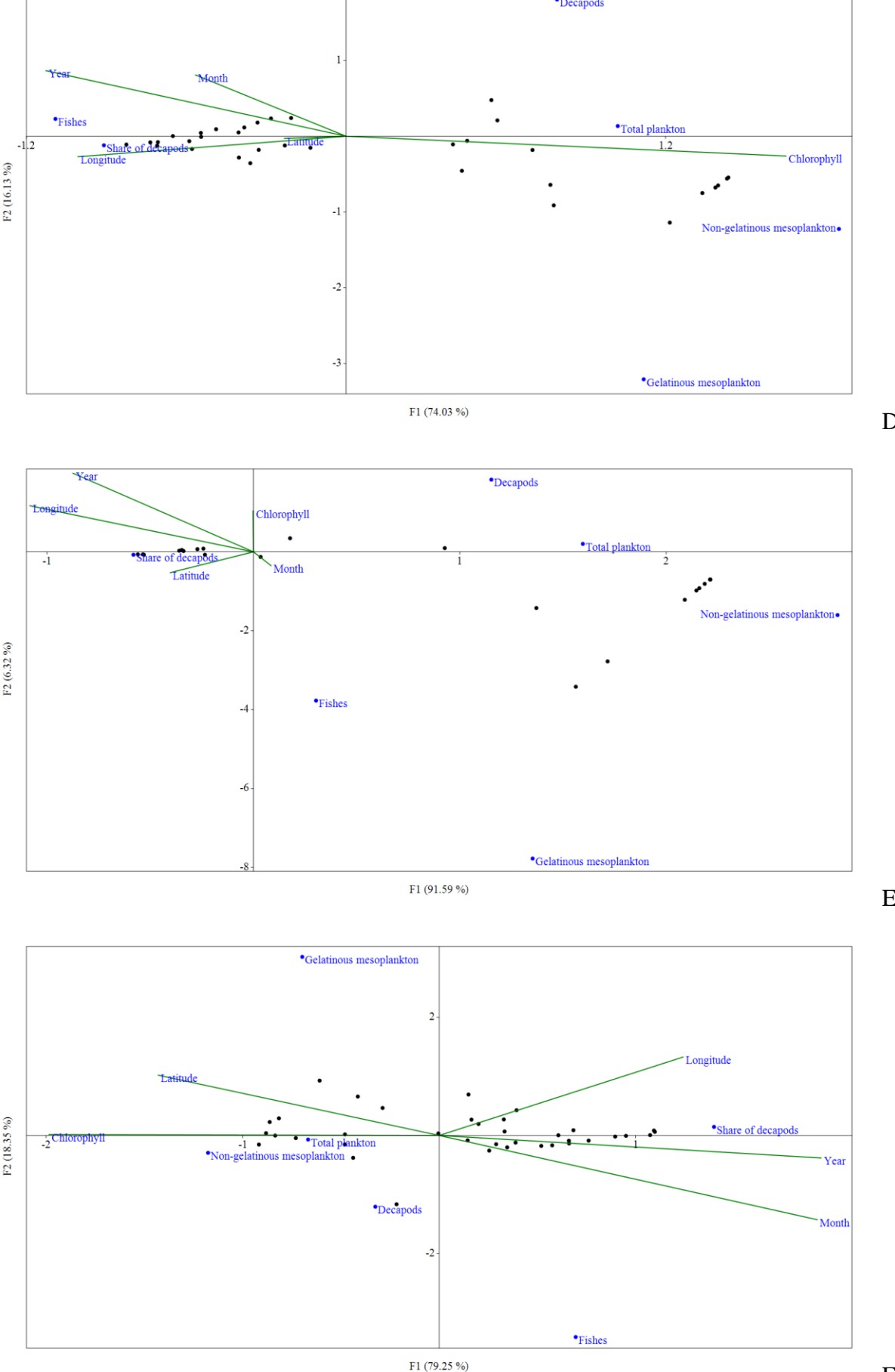

Figure 4.

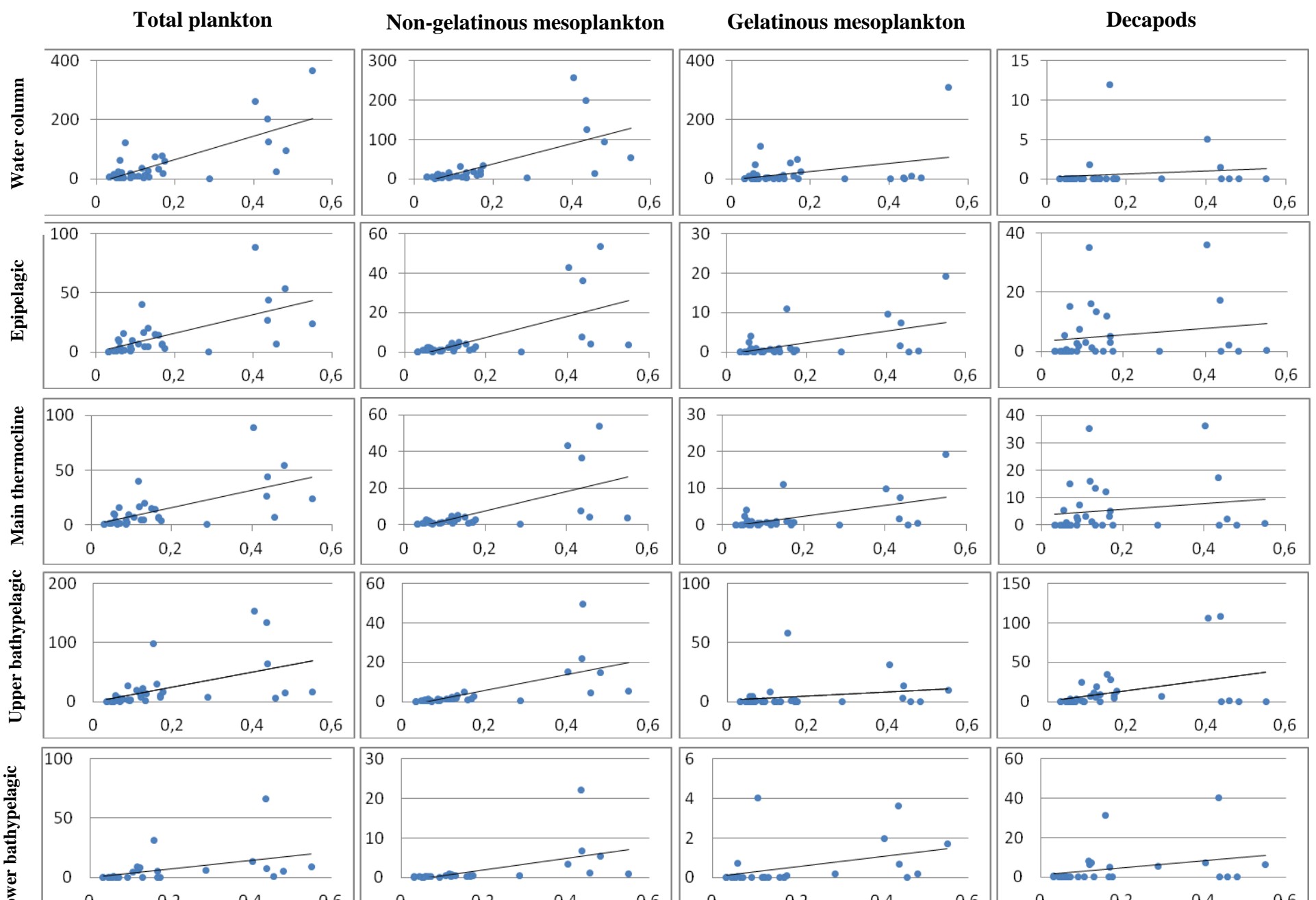

Figure 5.

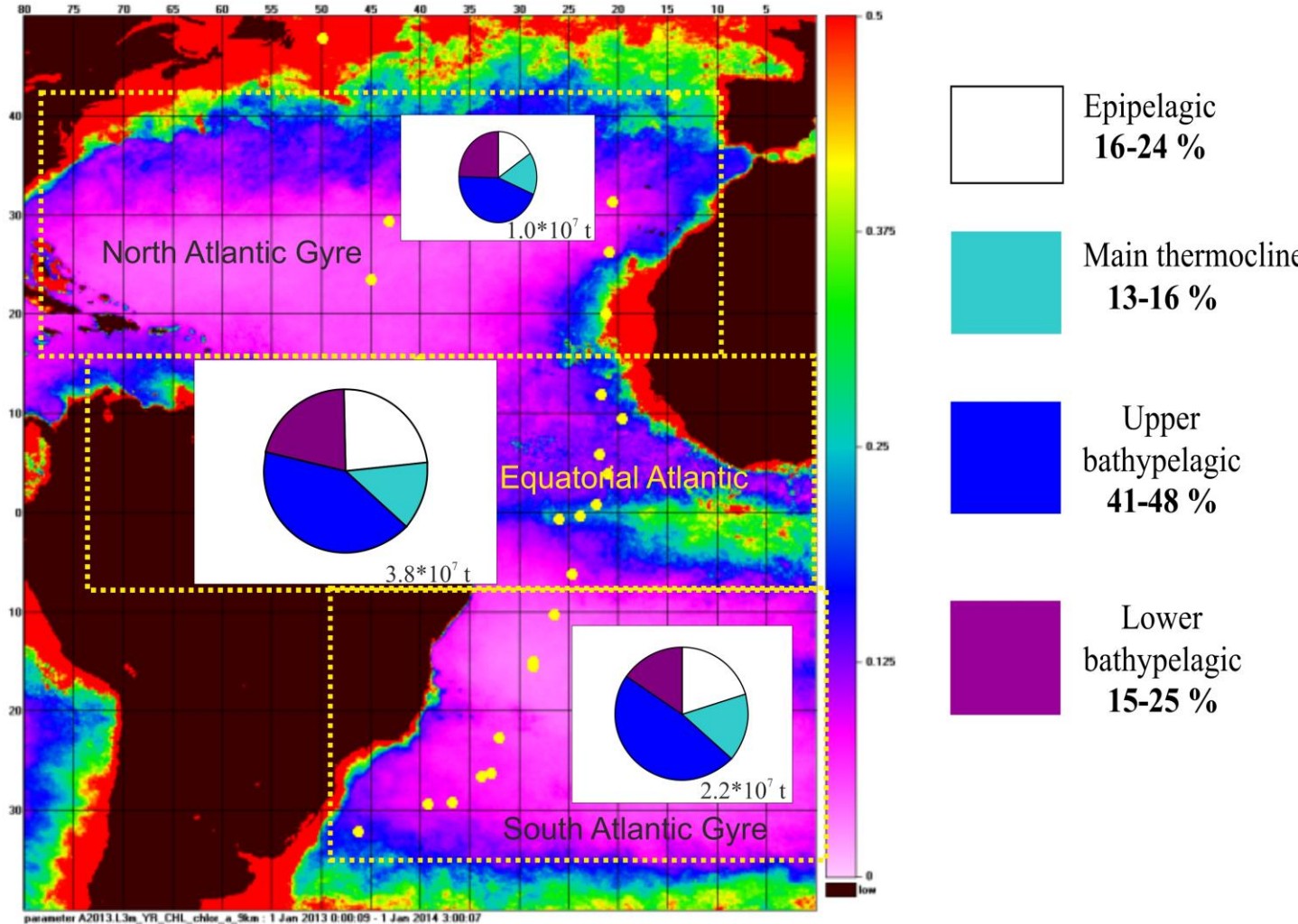

Figure 6.