# Peer review of "A NOVEL APPROACH REVEALS HIGH ZOOPLANKTON STANDING STOCK DEEP IN"

_Biogeosciences, 2016_

## Referee Comment (RC1) · A. Yamaguchi (Referee) · 18 Jun 2016

This manuscript provides biomass and taxonomic compositions of meso- and macro-zooplankton down to greater depths. Correlation analyses between surface chlorophyll a and standing stocks of various large-sized zooplankton are quite interesting and in-cluding important issues. Dominance of shrimp in deep-layer is not fully documented previously. This study showed their importance clearly. The message of this study is clear, robust and easy to understand. Following my comment is very minor.

My only concern is at Discussion on net avoidance. For me, mentioning that vertical tow is more robust for net avoidance than that of the horizontal tow is anomalous. While discussion on this subject (net avoidance of shrimp) is not at main focus of this study, their effect may have large impact for implication of this study.

---

## Referee Comment (RC2) · Anonymous Referee #2 · 28 Jul 2016

This manuscript presents an interesting attempt trying to explain deep sea zooplankton biomass with surface chlorophyll concentrations. The importance of the conclusion is clear as the ability to predict deep-sea zooplankton biomass from satellite-derived proxies would be very useful. However, the results are not fully presented and cast doubt on the overall conclusion.

In particular: - what is the actual data coverage? - how many nets were deployed in total? - how many organisms were counted? - what seasons have been sampled? - what are the uncertainties on your estimates?

A detailed list and some metadata analysis (beyond Fig 1 and 2) would be useful here. Maybe a table in the supplementary material (e.g. with sampling date, time and location, temperature, surface Chl, volume samples, number of individuals counted)?

Also, a figure similar to Fig 2 but with biomass instead would be useful.

A scatter plot of the raw data (biomass vs Chl) is needed to properly evaluate the correlations summarized in Table 2.

Over 300 taxa were identified but the data was then combined to three vaguely defined groups. I am surprised that community composition is not presented, and I think it would greatly enhance the manuscript.

Throughout the manuscript, Chl concentrations and primary production are used synonymously. They are not the same and should therefore be discussed more carefully (e.g. L53).

L120ff: Is this contribution normalized for the depth horizons? Otherwise it is not surprising that the integrated biomass in the bathypelagic (1500 to 3000 m depth) is larger than in the mesopelagic (200 to ∼600 m depth). Indeed Table 1 shows that biomass per m3 was highest in the mesopelagic. This difference and the implications are worthwhile discussing.

Potential time lags between surface chlorophyll concentrations and biomass in the deep sea have not been discussed.

You do not explain why you excluded data from temperate waters (L74). A justification is needed.

The introduction should give a fairer representation of the literature. It is, for example, not true that information about deep-sea zooplankton is available for the North Atlantic only (L37).

Minor comments: Throughout the manuscript, it would be useful if the type of biomass that is referred to would be made clear (e.g. Table referring to wet weight?).

You used a 500-$\mu$m net (L82) but only present data on zooplankton >1 mm (L60). Why is this?

Overall, this study is very interesting and the results could be important. However, the data presentation and discussion need work to give the overall conclusion credibility.

---

## Author Comment (AC1) · 17 Aug 2016

Dear Editor and Referees, Attached is a new version of the Ms, all new/significantly corrected parts are in blue/red. See detailed comments below.

Yamaguchi (Referee) a-yama@fish.hokudai.ac.jp This manuscript provides biomass and taxonomic compositions of meso- and macrozooplankton down to greater depths. Correlation analyses between surface chlorophyll a and standing stocks of various large-sized zooplankton are quite interesting and including important issues. Dominance of shrimp in deep-layer is not fully documented previously. This study showed their importance clearly. The message of this study is clear, robust and easy to understand. Following my comment is very minor.

My only concern is at Discussion on net avoidance. For me, mentioning that vertical tow is more robust for net avoidance than that of the horizontal tow is anomalous. While discussion on this subject (net avoidance of shrimp) is not at main focus of this study, their effect may have large impact for implication of this study. Authors' comment 1: We thank Dr. Yamaguchi for his comments and for a high evaluation of our work. Concerning the net avoidance, we just try to explain why vertical hauls give evidence for higher SHRIMP (only shrimp) biomass values (respective to the horizontal hauls). We feel that our explanations based on visual observations may make sense. As a possible compromise, we have removed the last sentence "We propose the use of vertically, not horizontally, towed large nets for more representative assessments of deep-pelagic shrimp abundance". We have also added the following paragraph: "In contrast to decapods, pelagic fishes escape in horizontal direction, as has been observed from submersibles many times by the authors. This reaction is successful when vertical hauls are used and our results are thus not representative for assessment of the pelagic fish biomass. This biomass may occur to be finally correlated with Chl but horizontally towed gears are necessary to prove that".
In particular: - what is the actual data coverage? - how many nets were deployed in total? - how many organisms were counted? - what seasons have been sampled? - what are the uncertainties on your estimates? A detailed list and some metadata analysis (beyond Fig 1 and 2) would be useful here. Maybe a table in the supplementary material (e.g. with sampling date, time and location, temperature, surface Chl,

volume samples, number of individuals counted)? Authors' comment 2: A new Table 1 including additional information about samples is now included.

Also, a figure similar to Fig 2 but with biomass instead would be useful. Authors' comment 3: This figure may be given for a certain transect only, not for the whole dataset discussed. Part of data along a submeridional transect (36th and 37th Cruises of the R/V " Akademik Sergey Vavilov ") has been analyzed in detail and published with transect in a more specialized journal; see also comment 5.

A scatter plot of the raw data (biomass vs Chl) is needed to properly evaluate the correlations summarized in Table 2. Authors' comment 4: Done as a new figure 5. In addition, Table 3 is redone to represent more information about regressions (coefficients of determination ($R^2$), equations, and levels of significance).

Over 300 taxa were identified but the data was then combined to three vaguely defined groups. I am surprised that community composition is not presented, and I think it would greatly enhance the manuscript. Authors' comment 5: In fact, the whole database of this work contains two different datasets: (1) data of 2012-2013 (R/V "Akademik Sergey Vavilov", mainly Central and South Atlantic) and (2) data of 1994-2005 (R/V "Akademik Mstislav Keldysh", mainly North Atlantic). Samples have been taken with the same protocol, but identification was much more precise for the first dataset. The community composition, diversity, and other community patterns have been analyzed in detail for the first dataset and presented in a recent paper (A Vereshchaka, G Abyzova, A Lunina, E Musaeva, 2016. The deep-sea zooplankton of the North, Central, and South Atlantic: Biomass, abundance, diversity. Deep Sea Research Part II: Topical Studies in Oceanography. DOI: 10.1016/j.dsr2.2016.06.017). The second dataset contains representative biomass values and significantly contributes to the metadata concerning deep zooplankton; here we combine both datasets for a more comprehensive analysis. Now we clarify the situation in the Methods section.

Throughout the manuscript, Chl concentrations and primary production are used synonymously. They are not the same and should therefore be discussed more carefully (e.g. L53). Authors' comment 6: Corrected

L120ff: Is this contribution normalized for the depth horizons? Otherwise it is not surprising that the integrated biomass in the bathypelagic (1500 to 3000 m depth) is larger than in the mesopelagic (200 to âĹij600 m depth). Indeed Table 1 shows that biomass per m3 was highest in the upper bathy-pelagic. This difference and the implications are worthwhile discussing. Authors' comment 7: No, contribution is not normalized for the depth horizons, as we clearly consider standing stocks. The contribution of different depth layers is not obvious: could the anonymous reviewer foresee that the standing stock in the upper bathypelagic (ca. 700 m thick) is much higher than in the more rich mesopelagic (ca. 400 m thick) and even more productive epipelagic (200 m thick)? By the way, Table 1 shows that biomass per m3 was highest in the epipelagic followed by the upper bathypelagic (NOT mesopelagic) for the total plankton and most groups.

Potential time lags between surface chlorophyll concentrations and biomass in the deep sea have not been discussed. You do not explain why you excluded data from temperate waters (L74). A justification is needed. Authors' comment 8: We have added in the "Method" chapter the following sentence: "We excluded data from temperate waters where the major spring peaks in primary production are being exported from the euphotic zone (0–200 m depth) and reaching abyssal depths (4000 m) with a significant time lag (e.g., 42 days: Smith et al., 2002); this lag differs for different depth zones that may corrupt possible correlations".

The introduction should give a fairer representation of the literature. It is, for example, not true that information about deep-sea zooplankton is available for the North Atlantic only (L37). Authors' comment 9: L37 is an awkward construction, which is now corrected and a greater reference list across all Oceans is provided in the Introduction; for example, the following sentence is included: "Studies on the deep-sea plankton biomass at selected sites include those in the North Pacific (e.g., Vinogradov,

1968; Murano et al., 1976; Yamaguchi et al., 2002a, b; Yamaguchi, 2004) and Eastern Tropical Pacific (Sameoto, 1986), North Atlantic (Koppelmann and Weikert, 1992; 1999; Gislason, 2003; Vinogradov, 2005) and Mediterranean Sea (Scotto di Carlo et al., 1984; Weikert and Trinkaus, 1990), Indian Ocean (Vinogradov, 1968) and Arabian Sea (Koppelmann and Weikert, 1992; Böttger-Schnack, 1996)".

Minor comments: Throughout the manuscript, it would be useful if the type of biomass that is referred to would be made clear (e.g. Table referring to wet weight?). Authors' comment 10: Corrected throughout the text, in tables, and figure captions.

You used a 500-$\mu$m net (L82) but only present data on zooplankton >1 mm (L60). Why is this? Authors' comment 11: Now we have added in Methods: "We used a closing Bogorov-Rass (BR) plankton net (1-m2 opening, 500-$\mu$m mesh size, towed at a speed of 1 m sec-1), which was proven to successfully sample deep-sea plankton $\geq$ 1.0 mm long (Vinogradov et al., 1996; 2000); smaller animals may pass through the sieve during filtration."

Overall, this study is very interesting and the results could be important. However, the data presentation and discussion need work to give the overall conclusion credibility.

ADDITIONAL CONRIBUTION 1: Now we used Canonical Correspondence Analysis (CCA: Ter Braak, 1986) to quantify the following environmental variables: month and year as possible temporal explanatory variables, latitude, longitude, and depth as possible spatial explanatory variables, and chlorophyll. As the sampling was associated with distinct water masses, such environmental parameters as temperature, salinity, and depth were correlated; only one and simplest of them, the depth, was included in CCAs. CCAs have shown that depth and averaged surface chlorophyll have major effect on the group biomass and predict plankton biomass better. It is only after CCA that we use further correlation analyses.

ADDITIONAL CONRIBUTION 2: We have also provided six actual biomass profiles from the surface to the bathypelagic zone for several distant sites (Fig. 2).

Please also note the supplement to this comment:
http://www.biogeosciences-discuss.net/bg-2016-145/bg-2016-145-AC1-
supplement.pdf

―――――――――――――――――

[Figure]

**Supplement:**

[revised manuscript text omitted]

Non-gelatinous mesoplankton
Gelatinous mesoplankton
Fishes
Decapods

Figure 3.

[Figure]

A

[Figure]

B

[Figure]

C

[Figure]

Figure 4.

[Figure]

Figure 5.

[Figure]

Figure 6.

---

## Author Response (AR2)

Dear Editor and Reviewer,
Attached is a second major revision. Like before, all corrections are in blue. Detailed comments
are below.
I find that the manuscript has improved a lot. Thank you for following my comments and
suggestions. Unfortunately, I still think that the manuscript needs some work, particularly
regarding the statistics and figures. Please find my comments below.
AUTHORS: Dear referee, it's a pleasure to read these lines.
Major comments:
Thank you for showing the scatter plots with the raw data. This is very useful in determining
whether fitting a linear regression to your data is sensible. It looks like your data is significantly
right-skewed: the majority of your data points are clustered near the origin and the trend you
observe is driven by a few individual points. Please check whether your data is normally
distributed before fitting a linear regression. A log-log transformation might help to fulfil the
assumptions (i.e. normal distribution). You will likely still obtain a significant relationship
between Chl and zooplankton abundance after transforming the data, but the results will be
robust.
AUTHORS: Yes, most of tests for normal distribution failed. The data were log-transformed as
recommended and all distributions except a single one became normal. Figure 5, Table 3, and a
respective part in Results were redone.
The notion of an inverted food pyramid is very interesting and it would be nice to have a
graphical representation of this. Have you considered, maybe for a future study, to look at
biomass spectra?
AUTHORS: Yes, we plan to do this in the nearest future after collecting additional material this
year. We feel this idea needs a separate detailed paper with results and discussion. We would be
pleased to send this paper to you for a review next year.
All figures need work. Many are lacking axis labels and legends.
AUTHORS: You could overlook legends, they were submitted and stand separately of the
figures themselves. All captions include necessary (and not redundant) information including
information about axes (it would not be wise, for example, to label all identical axes in Fig. 5,
this information is included in the caption to this figure). If we have overlooked something,
please, please indicate more specifically.
The font sizes are often too small. Many captions are incomplete.
AUTHORS: You probably mean Fig. 4. We have enlarged the text. In Fig 5, dots look as if they
have different size; that is not true: this is the effect of our Word version, we will discuss with
the production department the picture format and proceed accordingly.
A point should be used to indicate decimal places (please also check the tables for this).
AUTHORS: Yes, the commas to indicate decimal places are now replaced with the points in Fig.
5 and Table 3. It is our fault, because traditional Russian format uses commas.
Minor comments:
Table 1: A table detailing the site of the locations and depth range has now been added. Why did
you not include Chl concentrations, temperature? This would allow other scientists to build on
your data.

AUTHORS: Now the required information added.

L22: "I doing so". Please change.

AUTHORS: removed

L25: Fish are not plankton, so I do not think you need to specifically mention them. Also, you exclude them from most of your data analyses.

AUTHORS: We have removed most of references except few ones, which are necessary for a general discussion.

L27: This sentence is a bit awkward as it mixes two ideas: vertical structure and trophic structure. I suggest rephrasing it to avoid confusion.

AUTHORS: corrected

L28: You are not discussion biogeochemical cycles. Maybe better: "These findings, […], suggest that the importance of deep-ocean pelagic fauna for biogeochemical cycles maybe more important than previously thought", or similar. Also, biogeochemical cycles are not mentioned anywhere else in the manuscript (except for in the abstract)!

AUTHORS: done as recommended

L80: I would move this paragraph into the method section.

AUTHORS: done as recommended

L98: "Samples have been taken following the same protocol"

AUTHORS: corrected

L120: What is the upper size that is reliably caught with these nets? This should be mentioned as it especially important for the interpretation of the fish biomass.

AUTHORS: size interval now provided

L127ff: maybe nicer to say "dominated by…" rather than "mainly…"

AUTHORS: done as recommended

L137: How did you obtain the length of an individual specimen?

AUTHORS: measured with an ocular ruler, now explained

L163: "decapod decapods", please correct

AUTHORS: sorry, corrected

L240: "quasiexponential decrease". Is this just based on eye-balling? Please clarify. A simple regression fit (i.e. biomass vs average depth) would make this statement more robust.

AUTHORS: We have removed "quasiexponential" for clarity. Indeed, more date are necessary to obtain robust exponential regressions. This is a task for the next two years. The cited authors (Vinogradov, 1970) actually had had exponential regressions on a more extensive material.

L243-245: This sentence is awkward and does not add much. Please rephrase or delete.

AUTHORS: deleted

L256: It is interesting that you decided to average Chl over one year. Out of interest, have you tried correlations between biomass and Chl averaged over, for example, 6 months or 1 month prior to sampling? I suspect that the correlation would be a lot weaker (if any).

AUTHORS: A very intriguing question. As mentioned above, this year we plan to get an
additional material (actually, to double material) and create a separate paper testing different
time and space averaging. The cruises start very soon and last long.